# Spontaneous Coronary Artery Dissection as a Cause of Acute Myocardial Infarction in COVID-19 Patients: A Case Report and Review of the Literature

**DOI:** 10.3390/healthcare12020214

**Published:** 2024-01-16

**Authors:** Angeliki Papageorgiou, Athanasios Moulias, Athanasios Papageorgiou, Theodoros Karampitsakos, Anastasios Apostolos, Grigorios Tsigkas, Periklis Davlouros

**Affiliations:** 1Cardiology Department, University Hospital of Patras, 26504 Patras, Greece; amoulias@upatras.gr (A.M.); or cardiology@pgnp.gr (A.P.); gregtsig@upatras.gr (G.T.); pdav@upatras.gr (P.D.); 2Ubben Center and Laboratory for Pulmonary Fibrosis Research, University of South Florida, Tampa, FL 33620, USA; tkarampitsakos@usf.edu; 3First Department of Cardiology, Hippokration General Hospital, 11527 Athens, Greece; anastasisapostolos@gmail.com

**Keywords:** spontaneous coronary artery dissection, non-atherosclerotic acute myocardial infarction, COVID-19 cardiac complications, causes of STEMI in COVID-19 patients

## Abstract

Patients with COVID-19 often experience significant cardiovascular complications, including heart failure, myocarditis, and acute coronary syndrome. We present the case of a male patient with severe COVID-19 pneumonia, complicated with inferior ST-segment elevation myocardial infarction (STEMI), which was attributed to spontaneous coronary artery dissection (SCAD). We also make a review of the literature on case reports of patients with COVID-19 and acute myocardial infarction due to SCAD. Through these clinical cases, a potential correlation between SCAD and COVID-19 infection is implied. Endothelial dysfunction, thrombotic complications, and disturbance of the vascular tone are established COVID-19 sequelae, triggered either by direct viral injury or mediated by the cytokines’ storm. These abnormalities in the coronary vasculature and the vasa vasorum could result in SCAD. Moreover, disturbances of the vascular tone can cause coronary vasospasm, a reported precipitant of SCAD. Thus, SCAD should be considered in COVID-19 patients with acute coronary syndrome (ACS), and in the case of STEMI, an early angiographic evaluation, if feasible, should be performed rather than thrombolysis to avoid potential adverse events of the latter in the setting of SCAD.

## 1. Introduction

Since the outbreak of the COVID-19 pandemic, it has become evident that apart from respiratory manifestations, SARS-CoV-2 can cause serious and even lethal cardiac complications. Acute cardiac injury, encountered in 19.7–27.8% of hospitalized patients, is associated with greater disease severity and mortality [1,2,3], and the majority of the reported cases include heart failure, acute coronary syndrome (ACS), and myocarditis [4]. Here, we present the case of a male patient hospitalized with severe COVID-19 pneumonia who exhibited ST-segment elevation myocardial infarction (STEMI) attributed to spontaneous coronary artery dissection (SCAD). Endothelial dysfunction, thrombotic complications, and disturbances of the vascular tone are established COVID-19 sequelae, triggered either by direct viral injury or mediated by the cytokines’ storm. These abnormalities in the coronary vasculature and the vasa vasorum could result in SCAD. Our case report also highlights the consideration of the optimal antithrombotic treatment in SCAD, especially in the context of a prothrombotic condition like COVID-19. We also make a review of the literature on similar case reports and highlight the fact that COVID-19 infection may provide the pathophysiologic substrate for the occurrence of SCAD.

## 2. Case Presentation

A 51-year-old white male with a medical history of hypertension was admitted to our hospital’s COVID-19 clinic due to a recent onset of fever, cough, and respiratory distress along with positive real-time polymerase chain reaction assay for SARS-CoV-2 infection. He was initially administered with oxygen via high-flow-nasal-cannula, dexamethasone, remdesivir, tocilizumab bolus, and fondaparinux for thromboprophylaxis. On day 7, while in severe respiratory distress (Figure 1) supported with a continuous positive airway pressure helmet, the patient complained of retrosternal chest pain radiating in his left arm with concomitant diaphoresis. His blood pressure was 140/80 mmHg and his pulse was 55 beats per minute. Lung auscultation revealed rales in both lungs, while the rest of the physical exam was normal. An electrocardiogram (ECG) was immediately obtained, which demonstrated normal sinus rhythm and 2 mm ST-segment elevation in leads II, III, AVF, V5, and V6 along with ST-segment depression in leads V1–V2.

The patient was loaded with aspirin (ASA) 325 mg and ticagrelor 180 mg and transferred to the catheterization laboratory, where emergent coronary angiography was performed via right radial access. During contrast injection in the left coronary system, a long smooth 70–80% stenosis was revealed along the mid portion of the left circumflex artery (LCX) extending to the distal part of the vessel, including an obtuse marginal branch (Figure 2). Despite intracoronary administration of nitroglycerin, the narrowing along the vessels remained unchanged; hence, we proceeded with intracoronary imaging using optical coherent tomography that revealed SCAD (Figure 3). The remainder of the coronary tree was normal.

In the catheterization laboratory, the patient was asymptomatic with Thrombolysis In Myocardial Infarction (TIMI) III flow in LCX; hence, a conservative approach was adopted. Peak high-sensitivity troponin level was 35,200 pg/mL. During his hospitalization, the patient received ASA and bisoprolol and continued his former antihypertensive treatment with a combination of angiotensin II receptor blocker/calcium channel antagonist for efficient blood pressure control. Fondaparinux 2.5 mg/day was also administered subcutaneously to prevent thromboembolism, while we decided to discontinue the second antiplatelet. His echocardiogram revealed hypokinesia in the inferior and inferolateral wall with an estimated left ventricular ejection fraction of 50%. His pulmonary function gradually recovered, and he was discharged 10 days later with no further complications, with ASA and rivaroxaban 10 mg once a day for 35 days. He also continued bisoprolol and his former antihypertensive treatment, as both factors (b-blockers and appropriate arterial pressure control) have been shown to mitigate the risk of recurrent SCAD [5].

At the one-month follow-up, the patient’s health remained uncomplicated. The echocardiographic study demonstrated a full recovery of systolic function with no hypokinesia present. The repeat angiographic image was also significantly improved (Figure 4). Rivaroxaban was discontinued and the patient was advised to continue ASA for at least a year. He was also subjected to computed tomography angiography from the brain to the pelvis as part of the screening for extra-coronary arteriopathies (especially fibromuscular dysplasia), which did not reveal abnormal findings. Laboratory analyses after one month from the acute infection, including antinuclear antibodies, antibodies to β2-macroglobulin, and rheumatoid factor, were also negative, excluding the presence of chronic systemic inflammatory disease. The present case report follows the CARE case report guidelines (“see Appendix A”).

## 3. Discussion

Spontaneous coronary artery dissection is an important cause of ACS, which accounts for 1.7–4% of cases [6]. In the new era of intracoronary imaging, it has become feasible to recognize even ambiguous cases, and it seems to be more frequent than initially thought. It most commonly affects women ≤ 50 years old, lacking traditional cardiovascular risk factors [6]. SCAD is the result of disruption in the vessel wall layers in any epicardial coronary artery with the formation of intramural hematoma and is non-iatrogenic, nontraumatic, and nonatherosclerotic. On the contrary, it has been associated with pregnancy, multiparity, and inflammatory or vascular conditions, especially fibromuscular dysplasia, and in >50% of cases, a precipitating factor is reported, including intense exercise, Valsalva, emotional stress, coughing, vomiting, or labor and delivery [7]. The diagnosis is initially achieved by catheter-based coronary angiography with the implementation of intravascular imaging in challenging cases [6]. Cardiac magnetic resonance (CMR) has been used for the diagnosis of SCAD, and recently, there is evidence that it may recognize patients at high risk for adverse cardiac events [8]. The management is mainly conservative, as it has been shown that intervention in the labile vessel wall can result in the extension of the dissection and worse outcomes, except for patients with hemodynamic instability, left main artery dissection, or ongoing ischemia, where revascularization is the only viable option [6].

SARS-CoV-2 is a single-stranded RNA virus. Its entry into the human body is mediated mainly by the binding of the S1 unit of the spike protein to the angiotensin-converting enzyme-2 (ACE-2) receptor of the host cells, which is commonly encountered in the lungs, heart, and blood vessels. Despite direct damage to the host cells, SARS-CoV-2 promotes severe systemic inflammation and immune cell overactivation, leading to a “cytokine storm”. The imbalance of T-cell activation with dysregulated release of inflammatory molecules, including interleukin (IL)-6, IL-17, and other cytokines, plays a crucial role in the cardiovascular morbidity of COVID-19 [9].

### 3.1. Potential Pathophysiologic Mechanisms of SCAD Related to COVID-19

This clinical case, along with other case reports, suggests a possible association between SCAD and COVID-19 infection. It is well-established that patients with COVID-19 often experience significant cardiovascular complications, including heart failure, acute coronary syndromes, and myocarditis [4]. SCAD may be associated with various pathophysiologic mechanisms that occur during COVID-19 infection. Potential mechanisms include the following:The accumulation of macrophages and T-cells in the coronary adventitia and periadventitial fat found in autopsies of COVID-19 patients could induce excessive levels of cytokines and other mediators of the inflammatory response, resulting in disruption of the vessel wall layers and coronary dissection. Furthermore, the provoked sympathetic over-stimulation is an important factor in endothelial dysfunction and vulnerability [10].The infiltration of the vasa vasorum by the inflammatory cells may cause direct damage and ruptures with the formation of intramural hematoma. Apart from this, an additional factor contributing to the generation of intramural hematoma may be the excessive angiogenesis of the vasa vasorum owing to the cytokines and the other inflammatory molecules’ signaling [11].Except for the indirect damage mediated by the cytokines and the inflammatory cells, the SARS-CoV-2 virus binds directly to the ACE-2 receptors in the vascular endothelial and smooth cells and can cause inflammation in the coronary vessel wall, impairment of the vascular tone, and deregulation of the coagulation and fibrinolytic systems, making the vessel wall more prone to dissection [9].The dysregulation of the vascular tone can cause coronary artery spasm, an identified precipitant of SCAD [7].Finally, the administration of high doses of corticosteroids, commonly used in the treatment of COVID-19 patients, may also cause intimal tear of the weakened arterial wall, resulting in the formation of intramural hematoma [12].

### 3.2. Review of Case Reports on SCAD and COVID-19 Infection

Our search strategy, to identify relevant cases to our study, included querying PubMed, Scopus, and Google Scholar using the terms: “COVID-19 and spontaneous coronary dissection”, “COVID-19 and acute coronary syndromes”, and “COVID-19 and non-atherosclerotic causes of acute coronary syndromes”. Only cases that included diagnosed SCAD after or at the same time as COVID-19 infection were included (Table 1).

The incidence of males in the reported cases, who manifested SCAD after or during COVID-19 infection, was seven patients out of sixteen (44%), which is higher compared to the previous studies of patients with SCAD outside the COVID-19 context. In the typical epidemiological pattern of SCAD, 87–95% of cases predominantly occur in women aged between 44 and 53 years [6]. Regarding past medical history, most patients, *n* = 13 (81.2%), lacked the typical conditions or precipitating factors found in SCAD, including arteriopathies, connective tissue disorders, pregnancy, or multiparity [6]. Two of them (12.5%) were on corticosteroid treatment [12,13], one patient (6.25%) had migraines [14], and, in one case (6.25%), intense cough precipitated SCAD [15]. The timing of the occurrence of SCAD was synchronous with COVID-19 infection in 12 cases (75%) [14,15,16,17,18,19,20,21,22,23]; however, the remaining patients presented with SCAD after 2 weeks to 3 months of the illness. Notably, the severity of symptoms associated with the coronavirus infection appears to lack a direct correlation with the occurrence of SCAD. Among the documented cases, three patients remained asymptomatic (18.75%) [14,21,22] during the COVID-19 infection, while in eight patients (50%), only mild symptoms of the viral infection were evident. Furthermore, only five patients (31.25%) demonstrated severe disease [13,15,16,20].

Out of sixteen patients, six patients, including ours (37.5%), presented with STEMI [12,14,15,16,22,24], one with cardiogenic shock [20], and one survived cardiac arrest [25] owing to ventricular fibrillation. The remainder suffered from non-ST-segment elevation myocardial infarction (NSTEMI). The most affected vessel was LAD (62.5%), following LCX in four patients, RCA in three patients, and RI in one patient. Multiple coronary artery dissection was demonstrated in two patients (12.5%) [12,16]. In the literature, multivessel dissection is reported in 9–23% of cases [7]. All the patients survived, and the ejection fraction was restored within normal values in the cases where it is mentioned. Regarding the method of diagnosis, in all cases, coronary angiography was performed, except for the case of Ansari et al., in which CMR preceded coronary angiography [26]. Furthermore, in three cases, CMR was used as a follow-up method [20,23,24].

Great variability is shown regarding the antithrombotic treatment in patients with SCAD in general, and especially in patients with SCAD suffering simultaneously from COVID-19 infection. According to the existing data from these case reports, among patients that featured SCAD during the acute phase of the disease, two received DAPT following conservative treatment [14,16], one received coagulation on top of DAPT without any other obvious indications rather than COVID-19 infection [15], two were prescribed only ASA [17,20], two of them received DAPT following angioplasty [18,19], and one after plain old balloon angioplasty (POBA) was discharged on ASA alone [22]. Finally, one patient received clopidogrel plus warfarin in the presence of apical thrombus [23], and in our case, we decided on single antiplatelet therapy plus rivaroxaban in prophylactic dose to avoid thromboembolism during the acute phase.

### 3.3. STEMI Management during the COVID-19 Pandemic

At the outset of the COVID-19 pandemic, the first edition of the Consensus of Chinese Experts on diagnosis and treatment processes of acute myocardial infarction in the context of prevention and control of COVID-19 advocated that priority should be given to thrombolysis of ST-segment elevation myocardial infarction (STEMI) to mitigate the virus spreading [27]. This instruction was also adopted at first by some cardiologic communities in Europe. However, later, as the medical community gained more experience in the fight against COVID-19, the above directive was canceled and the European Society of Cardiology (ESC) published the document ESC guidance during the COVID-19 pandemic, which stated that: “The maximum delay from the STEMI diagnosis to the reperfusion of 120 min should remain the goal. Primary percutaneous coronary intervention remains the reperfusion therapy of choice, if feasible within this time frame and performed in facilities approved for the treatment of COVID-19 patients in a safe manner for healthcare providers and other patients” [9]. Thrombolytic therapy in SCAD may cause hematoma expansion and dissection progression, and in some case reports, is associated with adverse outcomes [6,28,29]. Notwithstanding, there is recent evidence from the Canadian SCAD study that thrombolysis may not lead to worse outcomes in STEMI patients with SCAD comparing patients who received thrombolysis with those who did not [30]. However, these findings must be interpreted with caution until solid evidence arises.

### 3.4. Optimal Antithrombotic Therapy in SCAD Patients with Concomitant COVID-19 Infection

Antithrombotic therapy is debatable in SCAD due to the lack of evidence from large, randomized trials. Currently, a proposed algorithm recommends DAPT for at least 2 to 4 weeks after SCAD and then continued low-dose ASA alone for a total of 3 to 12 months, encompassing the timeframe for SCAD healing [6]. Interestingly, there is an ongoing randomized clinical trial, examining the efficacy and safety of beta-blockers and dual antiplatelet agents (1 month versus 12 months) in patients with SCAD [31]. Anticoagulation once SCAD is diagnosed is usually discontinued due to theoretical concerns of extension of the dissection unless there is apparent intraluminal thrombus or other indications for systemic anticoagulation [6]. The coexistence of COVID-19 infection in our patient further complicated our decision regarding the antithrombotic strategy. We chose to administer a single antiplatelet agent (ASA) along with rivaroxaban 10 mg once daily for 35 days post-discharge. The rationale was the fact that our patient had an increased risk for thromboembolic events (IMPROVE score > 2). According to one randomized trial, in patients at high risk discharged after hospitalization due to COVID-19, thromboprophylaxis with rivaroxaban 10 mg/day for 35 days improved clinical outcomes compared with no extended thromboprophylaxis [32]. In our opinion, the increased risk of thromboembolism in our patient outweighed the potential risk of extending the dissection with the use of anticoagulants. In the case report of Papanikolaou et al., the authors also use DAPT plus anticoagulant, with no other apparent reason, and no adverse events are reported [15]. Furthermore, Kireev et al. treated their patient with DAPT and intravenous unfractionated heparin despite the presence of SCAD [12]. Finally, Pettinato et al. [13] and Bashir et al. [23], due to the occurrence of apical thrombus, used ASA, clopidogrel plus warfarin, and clopidogrel plus warfarin, respectively, with no adverse events reported. There is a lack of data regarding the best treatment after SCAD, which hence raises the need for future studies designed to investigate the optimal antithrombotic treatment after SCAD, especially in the context of thrombotic conditions, including COVID-19 infection.

## 4. Conclusions

Acute cardiac injury is commonly encountered in the setting of COVID-19 and is associated with greater disease severity and mortality [3]. The underlying cardiac pathology is widely variable, and SCAD may represent a cause. Thus, SCAD should be considered in COVID-19 patients with ACS, and in the case of STEMI, an early angiographic evaluation, if feasible, rather than thrombolysis is performed to avoid potential adverse events of the latter in the setting of SCAD. Currently, there are no clear guidelines on how to manage this subgroup of patients with SCAD and COVID-19, especially regarding antithrombotic treatment, considering the higher rate of thromboembolism in severely diseased COVID-19 patients. The more experience we gain, even from case reports, may contribute to this direction.

## Figures and Tables

**Figure 1 healthcare-12-00214-f001:**
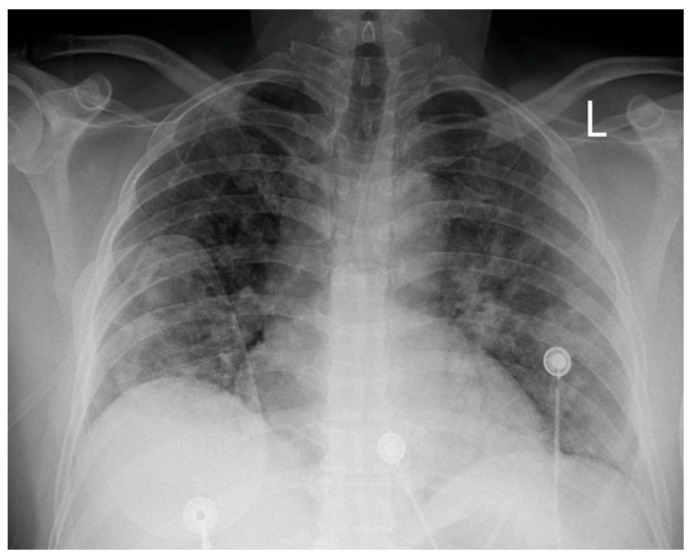
Chest X-ray on the day of the episode; (L: left side).

**Figure 2 healthcare-12-00214-f002:**
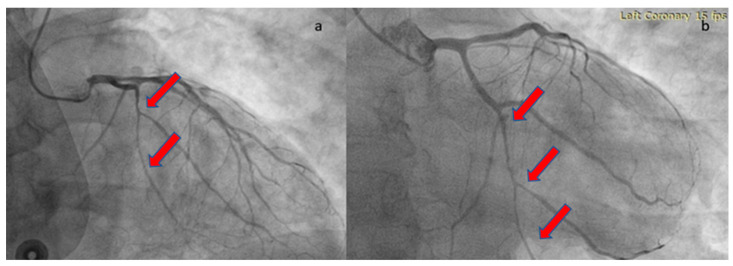
Coronary angiography revealing a long, smooth lesion in LCX. The red arrows show the long lesion. (**a**) RAO cranial angiographic view; (**b**) RAO caudal angiographic view.

**Figure 3 healthcare-12-00214-f003:**
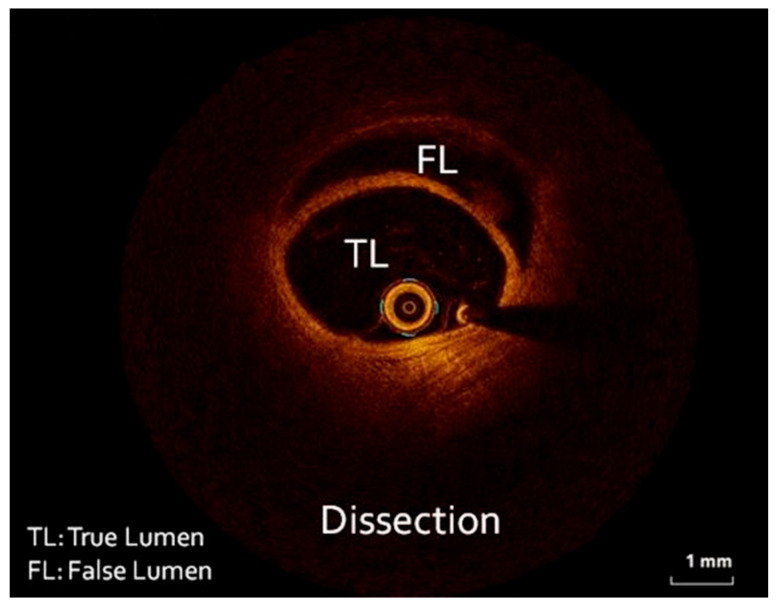
Optical coherent tomographic run depicting the true (TL) and the false lumen (FL) of the dissected vessel.

**Figure 4 healthcare-12-00214-f004:**
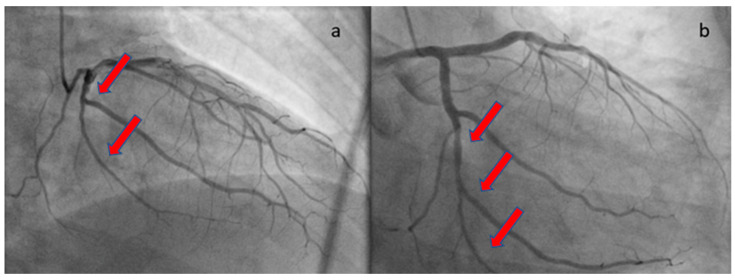
Repeat coronary angiography of the LCX, revealing angiographic healing. The red arrows show the improvement of the previously affected vessel sections. (**a**) RAO cranial angiographic view; (**b**) RAO caudal angiographic view.

**Table 1 healthcare-12-00214-t001:** Case reports concerning the occurrence of SCAD in patients infected with the coronavirus.

Author, Published Date	Sex, Age (Years)	Past Medical History	Timing of COVID-19 Infection, Regarding SCAD Occurrence	COVID-19 Severity	Diagnosis of ACS	Initial EF	Culprit Arteries	Treatment/Antithrombotics	Survival
Kireev et al., 2020 [12]	Male, 35	Serpinginous choroiditis (low CS dose), smoking, overweight	2 weeks ago	Mild	STEMI	N/A	RI 80%+ RCA 90%	PCI in RI+ conservative in RCA/DAPT+ anticoagulation (IV UFH)	Yes
Pettinato et al., 2022 [13]	Female, 43	MIS-A after COVID-19 (CS treatment), hypothyroidism	3 month ago	Mild at first—MIS-A later	NSTEMI	40%+ apical thrombus	LAD	Conservative/ASA+ clopidogrel+ warfarin for 1 month	Yes, with EF = 60%, thrombus resolution
Kumar et al., 2021 [14]	Female, 48	Migraines, hyperlipidemia	Index presentation	No symptoms	STEMI	45–50%	LAD	Conservative/DAPT for a year	Yes
Papanikolaou et al., 2020 [15]	Female, 51	HTN, smoking	Index hospitalization	Severe, high-flow-nasal-cannula	STEMI	N/A	LAD	Conservative/DAPT+ anticoagulation	Yes, with SCAD healing
Gasso et al., 2020 [16]	Male, 39	None	Index hospitalization	Severe/intubation	STEMI with no CV symptoms	50–55%, hypokinetic RV	LAD+ OM1	Conservative/DAPT	Yes
Courand et al., 2020 [17]	Male, 55	PAD	Index hospitalization	Mild	NSTEMI	60%	RCA	Conservative/ASA	Yes
Albiero et al., 2020 [18]	Male, 70	HTN, type 2 diabetes, PCI in LCX, smoking	Index hospitalization	Mild	NSTEMI	40–45%	LAD	PCI with DES/ASA+ clopidogrel	Yes, with EF improvement
Emren et al., 2021 [19]	Male, 50	None	Index presentation	Mild	STEMI	55%	RCA	PCI with BMS/ASA+ clopidogrel	Yes
Aparisi et al., 2021 [20]	Male, 40	None	Index hospitalization	Severe/intubation	Cardiogenic shock	35%	LAD	Conservative, ASA	Yes
Lewars et al., 2022 [21]	Female, 51	Anxiety, recovered postpartum CMP 15 years ago	At index presentation	No symptoms	NSTEMI	Dyskinetic apex, EF = 60%	LAD	Conservative/N/A	Yes, with SCAD healing
Shah et al., 2023 [22]	Female, 67	FHCAD	Index presentation	No symptoms	STEMI	65%, apical akinesia	LAD	POBA in LAD/ASA	Yes, recovery
Bashir et al., 2023 [23]	Female, 36	Morbid obesity (BMI = 49 kg/m^2^)	Index presentation	Mild, fever	NSTEMI	35%, thrombus formation	LAD 100%	Conservative/Clopidogrel+ warfarin	Yes, EF = 60%, thrombus resolution
Cannata et al., 2020 [24]	Female, 45	None	8 weeks ago, typical symptoms of COVID-19	Mild	STEMI	Anterior wall hypokinesia	LAD	Conservative/DAPT	Yes, with EF improvement
Ahmad et al., 2021 [25]	Female, 43	AF	12 weeks ago	Mild	Cardiogenic shock	20%	LCX	Conservative/Not reported	Yes, EF = 60%
Ansari et al., 2022 [26]	Female, 58	Hyperlipidemia	2 months ago	Mild lung involvement, severe thrombocytopenia	NSTEMI	50%	LCX	Conservative/ASA+ clopidogrel	Yes
Papageorgiou et al., 2023 (Index case)	Male, 51	HTN	Index presentation	Severe, continuous positive airway pressure-helmet	STEMI	50%	LCX 70%	Conservative/ASA for 1 year+ rivaroxaban 10 mg OD for 35 days	Yes, EF = 60%, SCAD healing

ACS: acute coronary syndrome; ASA: acetylsalicylic acid; BMI: body mass index; BMS: bare metal stent; CMP: cardiomyopathy; CS: corticosteroid; CV: cardiovascular; DAPT: dual antiplatelet therapy; DES: drug eluting stent; EF: ejection fraction; FHCAD: family history of coronary artery disease; HTN: hypertension; IV: intravenous; LAD: left anterior descending artery; LCX: left circumflex artery; MIC-A: multisystem inflammatory syndrome (adult); NSTEMI: Non ST-segment elevation myocardial infarction; OD: omni die; OM: obtuse marginal branch; PCI: percutaneous coronary intervention; POBA: plain old balloon angioplasty; RCA: right coronary artery; RI: ramus intermediate artery; SCAD: spontaneous coronary artery dissection; STEMI: ST-segment elevation myocardial infarction; UFH: unfractionated heparin.

## Data Availability

Data are contained within the article and Appendix A.

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
