# Peer review of "Spontaneous Coronary Artery Dissection as a Cause of Acute Myocardial Infarction in COVID-19 Patients: A Case Report and Review of the Literature"

_healthcare, 2024, doi:10.3390/healthcare12020214_

Round 1
Reviewer 1 Report
Comments and Suggestions for Authors
Papageorgiou et al. described an interesting case of myocardial infarction in a COVID patient caused by SCAD. In addition, they performed a complete review of the literature regarding similar cases.
The paper is well-written, the clinical case is accurate and well described and the review of the literature offers a interesting point of view of this clinical field.
I have some suggestions that could improve significantly the paper
1) Do the authors have information regarding the grade of coronary stenosis for the other patients?
2) Was CMR performed for the patient of the clinical case? It represents a usefull tool for diagnosis and follow up for SCAD
3) How many patients in your review of the literature performed CMR? Which were the findings reported?
3) I think it would be of great interest to implement the discussion with section regarding the role of CMR in the diagnostic work up for SCAD, especially when they cause AMI without obstructive coronary arteries. In case of MINOCA it gives crucial prognostic information (pleaese consider this interesting paper 10.1016/j.jcmg.2023.05.016)
Author Response
Dear Reviewer,
Thank you for giving us the opportunity to submit a revised draft of the case report and review of the literature entitled "Spontaneous coronary artery dissection as a cause of acute myocardial infarction in COVID-19 patients: a case report and review of the literature" which was submitted for publication in the Healthcare Journal.
We appreciate the time and effort that you dedicated to providing feedback on our manuscript, and we are grateful for your valuable comments.
We have incorporated most of the suggestions. Those changes are highlighted within the manuscript. Please see below, for a point-by-point response to your comments and concerns.
Reviewer #1
- Comment from Reviewer #1: Do the authors have information regarding the grade of coronary stenosis for the other patients?
Author’s response: We thank the reviewer for his/her comment. In only two more case reports, the grade of coronary stenosis is reported. I have filled in the severity of the stenosis of these cases in Table 1. The lack of this information in the majority of the cases may be attributed to the fact that SCAD type I has a more peculiar appearance that cannot be graded accurately, and type III lesion may be extended in the entire vessel length.
- Comment from Reviewer #1: Was CMR performed for the patient of the clinical case? It represents a useful tool for diagnosis and follow up for SCAD.
Author’s response: Our patient was not subjected to CMR. The diagnosis was achieved by coronary angiography and intracoronary imaging (OCT). At the follow-up visit coronary angiography was also performed. Although CMR is an undoubtfully useful tool for the diagnosis and mainly the follow-up of SCAD patients, unfortunately is not widely available in every hospital.
- Comment from Reviewer #1: How many patients in your review of the literature performed CMR? Which were the findings reported?
Author’s response: Four of the reported case reports performed CMR. In the case report of Cannata et al. CMR was done at the follow-up: “Cardiovascular magnetic resonance performed 3 weeks later, showed resolution of left ventricular function with evidence of infarction and elevated T2 in the mid- to apical septum and anterior walls.” The patient from the Aparisi et al. case report, performed a cardiac magnetic resonance at the follow-up, which revealed “discarded any signs of inflammation or fibrosis, a left ventricular ejection fraction of 35%, and resolution of the intraventricular thrombus.” The case report from Ansari et al., was subjected to CMR at the diagnosis and showed: “In T2-weighted fat suppression images of short tau inversion recovery sequences (STIR), there was evidence of subendocardial inflammation with linear hypointensity area in the mid to apical lateral. The late gadolinium enhancement images showed nearly transmural infarction in the mid to apical lateral with evidence of microvascular obstruction/intramyocardial hematoma hypointense core in an enhanced infarct core in line with STIR images and compatible with ischemic injury. There was also pericardial enhancement evident.” After that coronary angiography was performed, which revealed SCAD. Bashir et al., planned a CMR as a follow-up, but the results are not reported. We have briefly added the above in page 5, line 173-175).
- Comment from Reviewer #1: I think it would be of great interest to implement the discussion with section regarding the role of CMR in the diagnostic work up for SCAD, especially when they cause AMI without obstructive coronary arteries. In case of MINOCA it gives crucial prognostic information (please consider this interesting paper 10.1016/j.jcmg.2023.05.016)
Author’s response: We have added a brief comment in the discussion section regarding the prognostic role of CMR in page 4, line 109-113.
Reviewer 2 Report
Comments and Suggestions for Authors
Overall, the topic is interesting, relevant, and novel. The manuscript is objective, concise, and clear. The authors did a good job. My comments and suggestions aimed to help authors to reflect on issues that would improve this paper.
Major issues.
The authors did not follow the CARE guidelines. I strongly recommend the authors follow the CARE guidelines (https://www.equator-network.org/reporting-guidelines/care/) for completeness and supporting transparency of this manuscript. I suggest including the guidelines as an appendix in the manuscript.
The introduction should convince the reader that this case is unique and relevant. What is unique about this case? What is this case adding to the current literature? The authors need to articulate these points better in the introduction.
It needs to be clarified in the manuscript if the patient provided consent. Please include this information.
What are the limitations of this case study? Please include the limitations.
Minor issues
Section 3.2. Sometimes, the numbers and percentages are reported, sometimes only the numbers. I recommend being consistent when delivering the information. Numbers and percentages would be my suggestion.
Page 5, lines 149-150. "A substantial number of patients who manifested SCAD after or during COVID-19 149 infection were of male gender, specifically 7 patients out of 16. This sentence is confusing, I recommend the authors revise the sentence explaining that the incidence of males is higher compared to previous studies.
Author Response
Dear Reviewer,
Thank you for giving us the opportunity to submit a revised draft of the case report and review of the literature entitled "Spontaneous coronary artery dissection as a cause of acute myocardial infarction in COVID-19 patients: a case report and review of the literature" which was submitted for publication in the Healthcare Journal.
We appreciate the time and effort that you dedicated to providing feedback on our manuscript, and we are grateful for your valuable comments.
We have incorporated most of the suggestions. Those changes are highlighted within the manuscript. Please see below, for a point-by-point response to your comments and concerns.
Reviewer #2
- Comment from Reviewer #2: The authors did not follow the CARE guidelines. I strongly recommend the authors follow the CARE guidelines (https://www.equator-network.org/reporting-guidelines/care/) for completeness and supporting transparency of this manuscript. I suggest including the guidelines as an appendix in the manuscript.
Author’s response: We thank the reviewer for his/her comment. We have attached the CARE guidelines checklist below. We have also attached the timeline of our case report.
- Comment from Reviewer #2: The introduction should convince the reader that this case is unique and relevant. What is unique about this case? What is this case adding to the current literature? The authors need to articulate these points better in the introduction.
Author’s response: This has now been amended (page 1, line 38-43).
- Comment from Reviewer #2: It needs to be clarified in the manuscript if the patient provided consent. Please include this information.
Author’s response: We thank the reviewer for his/her comment. It is already stated in page 9, line 261-262.
- Comment from Reviewer #2: What are the limitations of this case study? Please include the limitations.
Author’s response: The main limitation of our study was the lack of data on the optimal antithrombotic treatment, which is thoroughly reported in page 8, line 229-245.
- Comment from Reviewer #2: Section 3.2. Sometimes, the numbers and percentages are reported, sometimes only the numbers. I recommend being consistent when delivering the information. Numbers and percentages would be my suggestion.
Author’s response: We have revised accordingly.
- Comment from Reviewer #2: Page 5, lines 149-150. "A substantial number of patients who manifested SCAD after or during COVID-19 149 infection were of male gender, specifically 7 patients out of 16. This sentence is confusing, I recommend the authors revise the sentence explaining that the incidence of males is higher compared to previous studies.
Author’s response: We have revised accordingly (page 5, line 154-156).
Timeline |
|
Day -7 |
Hospitalization due to COVID-19 pneumonia |
Day 0 |
Onset of acute chest pain |
|
Emergent coronary angiography with intracoronary imaging and SCAD diagnosis |
|
Conservative approach |
Day 10 |
Discharge from the hospital |
Day 30 |
Repeat coronary angiography as an outpatient with angiographic healing |

Reviewer 3 Report
Comments and Suggestions for Authors
The authors present an interesting and well-documented case report on a patient with SCAD presented as STEMI in the setting of acute COVID-19 infection. Also, they extensively discuss possible pathophysiologic mechanisms of SCAD related to COVID-19 infection. Finally, the authors present a comprehensive and up-to-date review of similar cases in the published literature focusing on the choice of antiplatelet/antithrombotic therapy. In the presented case, the decision for the single antiplatelet therapy plus rivaroxaban in prophylactic dose during the acute phase (first 35 days), followed only by ASA for at least one year (estimated timeframe for SCAD healing), seemed sound, well-documented and was shown to be effective finally.
The article is well written, and the enclosed literature is properly selected, presenting the current status of diagnostics and treatment of SCAD, with a particular accent on the acute COVID-19 infection environment. The article provides good and practical guidance to deal with similar clinical situations.
STUDY STRENGTHS:
The authors present an interesting and well-documented case report on a patient with SCAD presented as STEMI in the setting of acute COVID-19 infection. Also, they extensively discuss possible pathophysiologic mechanisms of SCAD related to COVID-19 infection. Finally, the authors present a comprehensive and up-to-date review of similar cases in the published literature focusing on the choice of antiplatelet/antithrombotic therapy. In the presented case, the decision for the single antiplatelet therapy plus rivaroxaban in prophylactic dose during the acute phase (first 35 days), followed only by ASA for at least one year (estimated timeframe for SCAD healing), seemed sound, well-documented and was shown to be effective finally.
The article is well written, and the enclosed literature is, to a large extent, appropriately selected, presenting the current status of diagnostics and treatment of SCAD, with a particular accent on the acute COVID-19 infection environment. The article provides valuable practical guidance to deal with similar clinical situations.
STUDY WEAKNESS - SUGGESTION TO THE AUTHORS FOR FURTHER IMPROVEMENT:
In the presented case, the authors did not use beta-blockers during the follow-up phase. Although there is a shred of conflicting evidence regarding the use of beta-blockers, the majority of literature supports their usage in SCAD scenario - unless contraindicated - based on their mechanisms of action that reduce the vessel shear stress (the approach extrapolated from the studies on aortic dissection), and supported by the recent meta-analysis on that matter (1). Noteworthily, there is an ongoing study on the use of beta-blockers and DAPT in the SCAD (2) setting, which is worth mentioning in the discussion since the expected results will shed more light on this area.
SUGGESTED LITERATURE TO IMPROVE DISCUSSION:
1) Chi G, Najafi H, Montazerin SM, Lee JJ. Factors associated with recurrent spontaneous coronary artery dissection: a systematic review and meta-analysis. Coron Artery Dis. (2022) 33(7):566–73.
2) Alfonso F, de la Torre Hernández JM, Ibáñez B, Sabaté M, Pan M, Gulati R, et al. Rationale and design of the BA-SCAD (beta-blockers and antiplatelet agents in patients with spontaneous coronary artery dissection) randomized clinical trial. Rev Esp Cardiol (Engl Ed). (2022) 75(6):515–22.
CONCLUSION:
Overall, the article merits publication following minor revisions that would include a discussion on using beta blockers in SCAD.
Author Response
Dear Reviewer,
Thank you for giving us the opportunity to submit a revised draft of the case report and review of the literature entitled "Spontaneous coronary artery dissection as a cause of acute myocardial infarction in COVID-19 patients: a case report and review of the literature" which was submitted for publication in the Healthcare Journal.
We appreciate the time and effort that you dedicated to providing feedback on our manuscript, and we are grateful for your valuable comments.
We have incorporated most of the suggestions. Those changes are highlighted within the manuscript. Please see below, for a point-by-point response to your comments and concerns.
Reviewer #3
- Comment from Reviewer #3: In the presented case, the authors did not use beta-blockers during the follow-up phase. Although there is a shred of conflicting evidence regarding the use of beta-blockers, the majority of literature supports their usage in SCAD scenario - unless contraindicated - based on their mechanisms of action that reduce the vessel shear stress (the approach extrapolated from the studies on aortic dissection) and supported by the recent meta-analysis on that matter (1). Noteworthily, there is an ongoing study on the use of beta-blockers and DAPT in the SCAD (2) setting, which is worth mentioning in the discussion since the expected results will shed more light on this area.
SUGGESTED LITERATURE TO IMPROVE DISCUSSION:
1) Chi G, Najafi H, Montazerin SM, Lee JJ. Factors associated with recurrent spontaneous coronary artery dissection: a systematic review and meta-analysis. Coron Artery Dis. (2022) 33(7):566–73.
2) Alfonso F, de la Torre Hernández JM, Ibáñez B, Sabaté M, Pan M, Gulati R, et al. Rationale and design of the BA-SCAD (beta-blockers and antiplatelet agents in patients with spontaneous coronary artery dissection) randomized clinical trial. Rev Esp Cardiol (Engl Ed). (2022) 75(6):515–22.
Author’s response: Thank you very much for the comment. Our patient received b- blocker at discharge, and he was advised to continue it lifelong. You are right that it is not clearly stated, so we added the comment in the respective section to avoid future readers misunderstanding (page 3, line 82). We also added a comment to the discussion section regarding BA-SCAD trial, as you proposed (page 8, line 217-219).
Round 2
Reviewer 1 Report
Comments and Suggestions for Authors
I thank the authors for the corrections that have substantially improved the paper.
Author Response
Dear Reviewer,
Thank you very much for the helpful comments.
Reviewer 2 Report
Comments and Suggestions for Authors
The authors did a good job answering and including the required information following the CARE guidelines. My only suggestion is to include the number of the page(s) in the CARE guidelines, not the check marks.
Author Response
Dear Reviewer,
Thank you very much for the helpful comments. The revised CARE list is here attached.